# Identifying Diurnal Variability of Brain Connectivity Patterns Using Graph Theory

**DOI:** 10.3390/brainsci11010111

**Published:** 2021-01-16

**Authors:** Farzad V. Farahani, Magdalena Fafrowicz, Waldemar Karwowski, Bartosz Bohaterewicz, Anna Maria Sobczak, Anna Ceglarek, Aleksandra Zyrkowska, Monika Ostrogorska, Barbara Sikora-Wachowicz, Koryna Lewandowska, Halszka Oginska, Anna Beres, Magdalena Hubalewska-Mazgaj, Tadeusz Marek

**Affiliations:** 1Computational Neuroergonomics Laboratory, Department of Industrial Engineering and Management Systems, University of Central Florida, Orlando, FL 32816, USA; wkar@ucf.edu; 2Department of Cognitive Neuroscience and Neuroergonomics, Institute of Applied Psychology, Jagiellonian University, 31-007 Kraków, Poland; bartosz.bohaterewicz@uj.edu.pl (B.B.); ann.marie.sobczak@gmail.com (A.M.S.); anna.ceglarek@uj.edu.pl (A.C.); aleksandra.zyrkowska@uj.edu.pl (A.Z.); barbara.wachowicz@uj.edu.pl (B.S.-W.); koryna.lewandowska@uj.edu.pl (K.L.); halszka.oginska@uj.edu.pl (H.O.); a.beres@uj.edu.pl (A.B.); marek@uj.edu.pl (T.M.); 3Malopolska Centre of Biotechnology, Jagiellonian University, 31-007 Kraków, Poland; 4Department of Psychology of Individual Differences, Psychological Diagnosis, and Psychometrics, Institute of Psychology, University of Social Sciences and Humanities, 03-815 Warsaw, Poland; 5Chair of Radiology, Medical College, Jagiellonian University, 31-007 Kraków, Poland; monika.cichocka@uj.edu.pl; 6Department of Drug Addiction Pharmacology, Maj Institute of Pharmacology, Polish Academy of Sciences, 01-224 Kraków, Poland; magdalena.hubalewska@uj.edu.pl

**Keywords:** brain connectivity, resting-state fMRI, circadian rhythm, chronotypes, graph theory

## Abstract

Significant differences exist in human brain functions affected by time of day and by people’s diurnal preferences (chronotypes) that are rarely considered in brain studies. In the current study, using network neuroscience and resting-state functional MRI (rs-fMRI) data, we examined the effect of both time of day and the individual’s chronotype on whole-brain network organization. In this regard, 62 participants (39 women; mean age: 23.97 ± 3.26 years; half morning- versus half evening-type) were scanned about 1 and 10 h after wake-up time for morning and evening sessions, respectively. We found evidence for a time-of-day effect on connectivity profiles but not for the effect of chronotype. Compared with the morning session, we found relatively higher small-worldness (an index that represents more efficient network organization) in the evening session, which suggests the dominance of sleep inertia over the circadian and homeostatic processes in the first hours after waking. Furthermore, local graph measures were changed, predominantly across the left hemisphere, in areas such as the precentral gyrus, putamen, inferior frontal gyrus (orbital part), inferior temporal gyrus, as well as the bilateral cerebellum. These findings show the variability of the functional neural network architecture during the day and improve our understanding of the role of time of day in resting-state functional networks.

## 1. Introduction

Most living organisms express a rhythmic cycle across a 24 h period (circadian rhythm) that controls several physiological processes such as sleep–wake patterns [1,2], metabolic activity [3], and body temperature [4], as well as various brain functions [5] such as attention [6], working memory [7], decision bias [8], motor [9], and visual detection [10] tasks.

As well as circadian rhythms, individuals have biologically different inclinations for when to sleep and when they are at their highest alertness and energy level, which are referred to as chronotypes [11]. Accordingly, people can be divided into morning-type (or early larks), evening-type (or night owls), and intermediate-type (or “neither-type”) [12]; the circadian typology moves toward later hours in night owls compared with early larks [13]. Chronotype differences have been reported to be highly influential to people’s cognition, behavior, and daily neural activity [1,5,12,14,15,16].

Although effects of circadian rhythms and chronotypes on whole-brain connectivity have been examined (some cases have only considered time of day [17]), the results are often contradictory and inconsistent. For example, a group of researchers believe that resting-state brain networks maintain a constant topological organization throughout the day [18,19], while others believe that brain networks, especially default mode, sensorimotor, and visual networks, show significant changes as the day progresses when we are at rest [20,21,22]. Additionally, Orban et al. [23], contrary to the common belief that “global brain signal is low in the morning and then increases in the midafternoon, and drops in the early evening”, showed that the global signal fluctuation is continuously decreasing during the day.

Utilizing a combination of graph-based knowledge and noninvasive imaging modality such as functional MRI (fMRI) helps to investigate (locally or globally) the brain functional connectivity at high temporal resolution [24,25,26,27,28,29]. In recent years, several studies have been conducted to identify topological changes in the brain networks that help us better understand the mechanisms underlying human cognition and neurological disorders [30,31,32,33,34,35,36,37,38,39]. For example, Lunsford-Avery et al. (2020) studied the relation between the regularity of sleep/wake patterns and brain connectome among adolescents and young adults to measure how these naturalistic patterns contribute to default mode network (DMN) topology [31]. In another study, Farahani et al. (2019) examined the effects of sleep restriction on the brain functional network and found significant topological alterations mostly across the limbic system, DMN, and visual network [33]. Disrupted brain network topology was examined in studies on patients with Parkinson’s disease [32], chronic insomnia [36], major depressive disorder [39], as well as preterm infants [35].

The main purpose of this study is to examine variations of neural activity at different times of the day in both chronotypes and detect circadian fluctuations of brain functional networks using rs-fMRI data. Based on our previous findings, we hypothesized that topological changes were mostly because of time of day rather than chronotype, and areas such as the default mode and sensorimotor networks underwent the most changes. To this end, we apply a graph-theoretic framework to extract the global and local changes in functional connectivity patterns and determine the informative regions that differ during the course of the day. Furthermore, we examine whether graph properties are correlated with the cognitive variables derived from the assessments and questionnaires across participants. The results provide a better understanding of the functional topology of the brain at rest over the course of the day.

## 2. Materials and Methods

### 2.1. Participants

Through online announcements, 5354 volunteers were selected to fill out the Chronotype Questionnaire [40] for assessing circadian preferences, the Epworth Sleepiness Scale (ESS) [41] for measuring daytime sleepiness, as well as a sleep–wake assessment. A total of 451 participants divided into morning (MT) or evening types (ET) were selected for PER3 VNTR polymorphism genotyping—only the subjects who were homozygous for the PER3 5/5 alleles (MT) and PER3 4/4 alleles (ET) were included. The procedure resulted in 73 healthy and young individuals in both chronotypes. Other selection criteria included age between 20 and 35 years, right-handedness (assessed by the Edinburgh Handedness Inventory [42]), no sleep deprivation, no neurological illness, and normal or corrected-to-normal vision. The selected individuals were scanned twice, first about 1 h (morning session) and then 10 h (evening session) after waking up. Session order was counterbalanced across participants. The final research sample for further analysis consisted of 62 subjects (39 women, mean age: 23.97 ± 3.26 years). Participants were asked to have adequate sleep for 1 week before the experiment, and actigraphs were used to monitor their sleep length and quality during that week as well as during the experiment days. Participants were prohibited from consuming alcohol (2 days) and caffeine (1 day) before the scanning sessions, and to refrain from strenuous activity during the experiment. The Ethics Committee for Biomedical Research at the Military Institute of Aviation Medicine, (Warsaw, Poland; 26 February 2013) and the Institute of Applied Psychology Ethics Committee of the Jagiellonian University (Kraków, Poland; 21 February 2017) approved the study, and an informed consent was obtained from all participants in accordance with the Declaration of Helsinki. Demographics, questionnaires, and actigraphy results are provided in Table 1.

### 2.2. Data Acquisition

Magnetic resonance imaging scans were performed using a 3T Siemens Skyra MRI system with a 64-channel head coil. Structural images were collected for each participant using a sagittal three-dimensional T1-weighted MPRAGE sequence. Functional resting-state blood oxygenation level-dependent (BOLD) signals were obtained through a gradient-echo single-short echo planar imaging sequence (10 min/run) using the following parameters: repetition time/echo time (TR/TE) = 1800/27 ms; field of view = 256 × 256 mm^2^; slice thickness = 4 mm (no gap); voxel size = 4 × 4 × 4 mm^3^. A total of 34 interleaved transverse slices and 335 volumes were obtained from selected participants. During the resting state, participants were instructed to lie in the scanner with their eyes open while thinking of nothing, and to remain awake throughout the scanning session. Participants’ awakeness was monitored by an eye tracking system (Eyelink 1000, SR Research, Mississauga, ON, Canada).

### 2.3. Data Preprocessing

Data were preprocessed using DPABI v. 4.2 and SPM 12, both working under Matlab v.2018a (The Mathworks Inc., Natick, MA, USA). To avoid instability of the initial MRI signals, the first 10 time points were discarded, and the data were then corrected for slice timing and head motion. Participants with movements in one or more of the orthogonal directions above 3 mm or rotation above 3° were excluded from the analysis; four participants were excluded due to excessive head movements. Subsequently, functional scans were coregistered using T1 images and normalized to the Montreal Neurological Institute (MNI) space using DARTEL [43] and a voxel size of 3 × 3 × 3 mm^3^. In total, seven participants were excluded because of the low quality of the normalization. The functional data were spatially smoothed with a 4 mm Full Width at Half Maximum (FWHM) kernel. The 24 motion parameters that were derived from the realignment step were regressed out from the functional data by linear regression, as well as five principal components from both cerebrospinal fluid and white matter signals using principal components analysis integrated in a component-based noise correction method [44]. The global signal was included because of its potential in providing additional valuable information [45], and the signal was band-pass filtered (0.01–0.1 Hz).

### 2.4. Network Construction and Analysis

In this study, we parcellated the whole brain into 116 distinct regions of interest (ROIs; 90 cortical and subcortical and 26 cerebellar) using the automated anatomical labeling (AAL) atlas [46]. The average time courses across all voxels within each region were extracted. Next, by means of Pearson’s correlation coefficients, we calculated the pairwise functional connectivity between ROIs. The results were transformed using Fisher’s r-to-z transformation for better normalization. Thus, a symmetrical adjacency matrix with a size of 116 × 116 was built for each participant (Figure 1). We applied a density-based thresholding on the obtained networks to maintain the strongest links and eliminate weaker ones [47]. The network density was set between 0.05 and 0.275 with a step of 0.025. Finally, we binarized the matrices to overcome the complexity issue.

### 2.5. Graph Measure Computation

We extracted a set of global and local properties of the binary networks for each participant using the Brain Connectivity Toolbox (BCT) [28]. Global properties such as global efficiency, modularity, average shortest path, small-worldness, and assortativity can be used to provide global information flow and functional specialization. Local properties such as degree, betweenness centrality, nodal efficiency, nodal clustering coefficient, and participant coefficient (for details on the measures see [28]), were computed for each region separately, reflecting the centrality of nodes and existence of hubs (connector or provincial) in the network. All measures were extracted from the thresholded and binarized networks with the sparsity between 0.05 and 0.275 (step of 0.025). The reason for choosing this interval was to reduce computational complexity while preventing the creation of disconnected graphs.

## 3. Results

### 3.1. Global Analysis

We found a significant increase in small-worldness (the ratio of normalized clustering coefficient to normalized path length) from the morning to the evening session (Figure 2) at higher densities (*p* < 0.01, Bonferroni corrected), whereas the changes were not significant in terms of chronotypes. No significant variations were observed for other global measures.

### 3.2. Local Analysis

Table 2 shows the results of the paired *t*-test on the brain regions that differed statistically between the morning and evening sessions using local metrics, including degree centrality, betweenness centrality, clustering coefficient, and nodal efficiency. According to Table 2, several meaningful changes were evident, mostly across the left hemisphere, in areas such as the precentral gyrus, orbital part of inferior frontal gyrus, lentiform nucleus (particularly the putamen), inferior temporal gyrus, and a series of regions inside the cerebellum. No significant differences were observed for other local measures (*p* > 0.001, Bonferroni corrected). Moreover, the results of degree centrality and betweenness centrality of all 116 brain areas are visualized in Figure 3. Compared with the morning session, the evening session showed significantly decreased degree centrality in the left precentral gyrus, the dorsolateral part of left superior frontal gyrus, the left supplementary motor area, the supramarginal and angular gyri of the left inferior parietal lobe, the left putamen, the left thalamus, and bilateral inferior temporal gyrus, and increased degree centrality in some areas within the cerebellum (*p* < 0.001, Bonferroni corrected).

### 3.3. Hub Analysis

We also identified network hubs along with their types (i.e., connector or provincial) in morning and evening sessions within the sensorimotor, visual, frontoparietal, default mode, limbic, and cerebellar networks (Table 3). The results are based on the mean connectivity matrix (across all participants for each corresponding session) and a network density of 0.1. According to Table 3, differences between the two sessions were located in regions such as the left supramarginal gyrus; right superior temporal pole; right thalamus; left lobule VIII of cerebellar hemisphere; and lobules IV, V, and VI of vermis. Interestingly, the sensorimotor network was the densest part of the brain at rest with the most hubs (mostly provincial, i.e., within modular connections) compared with the other networks. In contrast, the hubs in default mode, limbic, and cerebellar networks were mainly connector type (i.e., between modular connections).

### 3.4. Correlation Analysis

Finally, we performed correlation analyses to examine the associations between local measures and questionnaire scores (e.g., morningness/eveningness (ME) scale, amplitude (AM) scale, and ESS). The results are displayed in Table 4 (*p* < 0.01, Bonferroni corrected). For the morning session, we found significant negative associations between ME score and nodal properties of right hippocampus and right parahippocampal gyrus, as well as positive associations between ME score and nodal metrics (degree and nodal efficiency) of the right lenticular nucleus and pallidum. We found significant positive associations between AM score and nodal metrics (degree and nodal efficiency) of the left precentral gyrus and left postcentral gyrus, as well as negative associations between AM score and degree and betweenness centrality of the right lobule X of cerebellum. Finally, the only significant correlation with ESS score was its positive associations with degree and nodal efficiency of the left postcentral gyrus. In the evening session, we found significant negative associations between AM score and nodal metrics (nodal clustering coefficient and local efficiency) of the right hippocampus, as well as positive associations between ME score and degree centrality of the right pallidum. Furthermore, positive and negative correlations were observed for ME and AM, respectively, with nodal metrics within the left parahippocampal gyrus. No significant correlations were found between ESS and brain metrics.

## 4. Discussion

In this paper, we investigated the effect of time of day and the individual’s chronotype on the functional brain networks of 62 healthy participants using rs-fMRI data and a graph-based approach. In the global analysis, we found that small-worldness increased over the course of the day (*p* < 0.01, Bonferroni corrected). In the local analysis, we identified significant diurnal variations, mostly across the left hemisphere, in areas including the precentral gyrus, putamen, inferior frontal gyrus (orbital part), inferior temporal gyrus, as well as in the bilateral cerebellum (*p* < 0.001, Bonferroni corrected). In the hub analysis, we found that the sensorimotor network was the densest area of the brain (in terms of hub numbers) in both the morning and evening sessions with primarily provincial type hubs, whereas hubs in default mode, limbic, and cerebellar networks were mostly of the connector type. The effect of chronotype and interaction between time of day and chronotype (so-called synchrony effect) were not observed in global and local analyses, which is in line with our previous study [48]. The synchrony effect was confirmed in a variety of cognitive domains [5,49] and in task fMRI characterized by high complexity [15]. In relation to the resting-state data, some recent reports revealed the influence of chronotype on resting-state functional connectivity (with contradicting results) [50,51]; however, they did not confirm the synchrony effect. Our findings regarding global and local connectivity profiles indicate the variability of the brain’s functional organization between morning and evening resting-state sessions as a universal phenomenon, independent of circadian typology. Finally, in the correlation analysis, we found evidence of associations between questionnaire scores and local metrics in several regions in both sessions, mostly related to morning. In the following, we discuss in more detail the significant diurnal changes related to small-worldness, local characteristics, hub, and correlation analysis.

### 4.1. Diurnal Variations in Small-Worldness

A small-world network is an intermediary between random and regular networks, consisting of a large number of short-range connections together with a few long-range shortcuts [52]. Mathematically, small-world networks have a high clustering coefficient and short average path length, which makes them superior to other networks in terms of functional segregation and integration, respectively [28,53]. A higher small-worldness global property of brain networks has been shown in younger versus older individuals [54] and in healthy controls compared with patients with Alzheimer’s disease [55]. According to our rs-fMRI findings, a lower value of small-worldness in the morning compared with the evening reflects a less efficient functional topology and greater wiring cost. The results could be explained by an effect called “sleep inertia”, which is believed to be the third process (Process W) of sleep regulation together with circadian rhythm (Process C) and homeostatic process (Process S) [56]. It refers to the transitional state between sleep and wake, characterized by impaired performance and reduced vigilance in the minutes or even hours after waking up [57]. This conflicts with the common intuitive belief that in the morning hours, the brain is recovered after the full night of sleep and should work most effectively. The occurrence and length of sleep inertia depend on the individual and on the sleep stage in which waking occurred or on previous sleep deprivation [58,59]. However, the exact function and neurophysiological basis of sleep inertia are still not fully known (for a review see [56]). Vallat et al. (2019) suggest that this phenomenon is caused by the loss of functional brain network segregation from the default mode network, which is also observed during sleep and periods of elevated sleepiness. Then, a progressive restoration of the functional segregation of the brain networks is possibly responsible for sleep inertia dissipation after awakening [60].

In the present study, we found that the global small-worldness index was higher in the evening after the whole day of functioning, compared with the morning, regardless of the participant’s chronotype. Results on small-worldness of human brain networks in relation to time of day and participant fatigue level remain mixed and even contradictory. Our results are in line with observations made by Liu et al. (2014) [61], who found that small-worldness properties of resting-state networks in sleep-deprived individuals are higher than those in well-rested individuals. Researchers have interpreted this effect as an indicator of a compensatory reorganization of the human brain network under conditions of resource shortages. In the current study, participants had good quality and length of sleep the night before the experiment, which was confirmed by data obtained from their wrist actigraphs. However, these results can be interpreted as possibly related to the homeostatic process [62] that is in control of sleep regulation and accumulates during time spent awake.

### 4.2. Diurnal Variations in Nodal Properties

In this subsection, we discuss the topological changes of the brain regions across the day in detail. Our findings here are classified based on the predefined brain networks in this study, that is, default mode network, frontoparietal network, sensorimotor network, visual network, limbic system, and cerebellar network.

#### 4.2.1. Default Mode Network (DMN)

Our results showed that time of day affected degree, betweenness centrality, and nodal efficiency in the precentral, superior frontal, and middle temporal gyri. These results can be seen as a proof of DMN variability through the day. The DMN was initially presumed to be exceptionally active when the mind is not focused, being in a state of wakeful rest and wandering [63]. The DMN is thought to be implicated in various aspects of self-referential processing [64], such as thinking about ourselves, remembering the past, and making plans for the future [65], and it is sometimes referred to as an anti-task network because the DMN is deactivated during goal-oriented tasks [66,67]. Diurnal variation of DMN was also found in the study of Jiang et al. (2016) [17], which revealed increased regional homogeneity (ReHo) and amplitude of low-frequency fluctuations (ALFF) in the morning hours compared with the evening. Results of this study are congruent with ours, showing that the precentral gyrus, also known as the primary motor cortex, is more significant for the network in the morning hours. However, Jiang et al. (2016) [17] observed decreased ReHo and ALFF in the superior frontal gyrus in the morning resting-state procedure, whereas our results indicated higher nodal efficiency of the same region in the morning. The superior frontal gyrus is thought to be associated with higher cognitive functions; however, its contribution remains obscure [68]. Disagreement in current studies investigating circadian rhythms prompts further exploration of the aforementioned subject. A meta-analysis by Fusar-Poli et al. (2009) [69] found that increased activity of the middle temporal gyrus (MTG) was present when participants were presented with emotional faces. The MTG was also identified as being recruited in automatic semantic processing and being especially active during demanding task execution [70].

A recent study by Xu et al. (2019) shed some light on the functional complexity of the MTG [71]. These authors identified four sub-regions, each with different specialization in, among others, social cognition and semantic and language processing, demonstrating MTG involvement in many cognitive functions. Our results showed increased betweenness centrality in the right MTG during the morning session compared with the evening session. Higher values of betweenness centrality suggest that MTG as a node participates in a large number of shortest paths, being a hub-like node, such that, on average, more information will pass through MTG than other nodes inside a network.

#### 4.2.2. Frontoparietal Network (FPN)

Diurnal changes were also observed in regard to local properties of the FPN, the network involved in executive control [72]. Similar to that in other brain networks, these alterations were observed in the left hemisphere. First, the orbital part of the left inferior frontal gyrus showed decreased degree centrality and betweenness centrality in the evening compared with the morning session. Additionally, the left inferior parietal lobe showed an analogous pattern of diurnal differences, with lower degree centrality in the evening compared with the morning hours. Taken together, these results show that both the inferior frontal and inferior parietal lobes have fewer functional connections with other brain networks in the evening, and the inferior frontal gyrus also had fewer short paths, which may suggest that its role is less central to the network in the evening [28]. Importantly, part of the parietal lobe, the left angular gyrus, showed higher betweenness centrality in the evening than in the morning. This might suggest that whereas the role of inferior frontal gyrus is diminished in the evening, the role of left angular gyrus becomes more central to the network, because a higher fraction of short paths is typical for the bridging nodes [28]. The left inferior gyrus is linked, among others, to inhibitory control of responses [73], whereas the left inferior parietal cortex is linked to attention shifting and mediating attentional flexibility [74]. Accordingly, increased resting-state functional connectivity between the left angular gyrus and other brain regions has been linked to sustained attention deficits in patients with multiple sclerosis [75]. In addition, diurnal changes in other local property measures, such as nodal efficiency, were present in both the left inferior frontal and parietal regions, whereas the left superior temporal pole showed diurnal variations in clustering coefficient. These findings reveal diurnal variability in local integration within the neighborhood of the inferior frontal and parietal nodes, as well as changes in clustered connectivity, that is, in the interconnectedness of nodes within the neighborhood of the left superior temporal pole [28]. Taken together, our results suggest that the control processes mediated by the FPN are less efficient in the evening hours, especially in terms of inhibition. Consistently, time-of-day effects on the brain activity of the frontal and parietal regions and on related processes have been demonstrated in previous studies [76,77].

#### 4.2.3. Sensorimotor Network (SMN)

The SMN, involved in the processing of sensory information and motor reactions, has diurnal rhythmicity, as confirmed by several studies [20,78]. In the current study, two nodes that are part of the SMN—the supplementary motor area and supramarginal gyrus—had different degree centrality and nodal efficiency according to the time of day. A previous study revealed that the left supplementary motor area has increased functional connectivity in the evening hours [48], which indicates alterations in daily activity of the SMN. The results of the current study are in contrast to previous results, showing fewer connections coming out of this structure in the evening, according to the graph measures. In reference to the supramarginal gyrus, Song et al. (2018) demonstrated the synchrony effect, such that evening types showed higher activity during the evening session compared with morning types [79]. We did not observe this synchrony effect in any of the graph measures; however, the study of Song et al. (2018) [79] was conducted using task fMRI, not resting-state fMRI.

#### 4.2.4. Visual Network (VN)

The within-subject variability in VN is well known [80,81]; for example, sleep debt and self-reported “sleepiness” are positively correlated with functional connectivity in the VN [82,83]. However, VN changes regarding time of day have not been thoroughly examined. In our study, diurnal variability was found in this network, regardless of the participant’s chronotype. In the right inferior occipital gyrus (IOG.R), we found different betweenness centrality according to the time of day. In the left inferior temporal gyrus (ITG.L), alterations in degree centrality and nodal efficiency were noted. Diurnal variability in the right inferior temporal gyrus (ITG.R) involved all three factors. This means that particular nodes of visual network have fewer functional links and shortest paths to other nodes in the brain in the evening than in the morning. These results are consistent with previous studies, in which a decrease in resting-state functional connectivity was observed between regions of VN from morning to evening [17,48,83,84]. In contrast, Gratton et al. (2018) found no time-of-day effect on VN [85]. According to Cordani et al. (2018) [86], resting-state BOLD signal in the visual cortex increases significantly between 8:00 and 17:00, and then it decreases significantly at 20:00 and increases (but not significantly) again at 23:00. There is still no satisfying and clear explanation of this phenomenon of sensory processing within the circadian VN [87,88]. Cordani et al. (2018) suggested that the human visual cortex is modulated by daylight changes, with compensatory mechanisms at dawn and dusk [86].

#### 4.2.5. Limbic System (LS)

We found two subcortical areas, traditionally considered parts of the LS, that showed differences in local properties depending on the time of day: the putamen and the thalamus. The putamen (one of the basal nuclei) and the caudate nucleus compose the dorsal striatum. Primarily, the structure is thought to play an important role in movement preparation and execution and in learning [89]. In the context of circadian variability, changes in activation were shown mainly for the left putamen [90]. It has been reported that the putamen response to rewards is lower in the afternoon or early evening compared with that in the morning hours [91,92]. In line with those reports, our findings (i.e., differences in local connectivity indicators) implied that the left putamen is less functionally connected with other brain areas in the evening. The thalamus is seen as a hub that passes sensory and motor information between the cerebral cortex and subcortical areas while taking part in regulation of the sleep–wake cycle [93]. The paraventricular thalamus (PVT) is known to be especially important in this regulation because it is reciprocally connected to the suprachiasmatic nuclei (SCN) and receives photic and circadian timing information [94]. Additionally, the thalamus shows circadian rhythmicity [83]. Here, we found that compared with morning hours, degree centrality in the left thalamus was decreased in the evening. Interestingly, Muto et al. (2016) indicated that subcortical areas, including basal ganglia and the thalamus, exhibit circadian modulation that follows the melatonin profile [83].

#### 4.2.6. Cerebellar Network (CRB)

Graph analyses revealed diurnal differences in the CRB associated with higher measures of network centrality such as nodal efficiency and degree and betweenness centrality in the evening compared with the morning. These results indicate the high ability of bilateral Crus I and II but also left lobules VIIB, VIII, and X of the cerebellar hemisphere to transmit the information to other regions included in the CRB [95]. Dynamic interaction is related to greater efficiency and thereby better functioning of the whole cerebellum, which, apart from basic motor control such as voluntary limb movements, balance, and maintaining posture [96], is associated with the visual attention process and working memory [97]. Diurnal rhythmicity of resting-state cerebellar activity has not been sufficiently examined yet; however, a task study conducted by Bonzano et al. (2016) showed higher morning activity of the cerebellum during both actual and mental movement tasks [98], which is contradictory to our results, which revealed higher efficiency of the CRB during evening fMRI sessions. Sami et al. (2014) showed an association between memory consolidation and Crus II [99], whereas our results revealed larger centrality measures in the same area. Moreover, Tzvi et al. (2015) reported striatal–cerebellar networks to mediate consolidation in a motor learning task [100]. Because of the lack of knowledge on resting-state fMRI time-of-day differences in the CRB, there is a clear need for further investigation.

### 4.3. Provincial and Connector Hubs

Hubs, a set of highly interconnected brain areas [101], are a set of integrative nodes and have a key role in functional connectivity networks within the human brain [102]. They are involved in transmitting the information across different areas of the brain by incorporating parallel and distributed networks [103] and have a key role in network organization [104]. In the present study, we tested participants twice a day, in the morning and in the evening, to identify the brain hubs under the resting state conditions within both experimental sessions. The analysis recognized the common—for morning and evening—provincial hubs (i.e., within modular connections) as a bilateral rolandic operculum, insula, postcentral gyrus, superior temporal gyrus, lingual gyrus fusiform gyrus, precentral gyrus, midcingulate area, and cerebellum. We also found the common—for both testing sessions—connector hubs (i.e., between modular connections) to be bilateral precentra gyrus, midcingulate area, and lobule VI of the cerebellar hemisphere. The differences between the two experimental sessions were located in regions such as the left supramarginal gyrus; right superior temporal pole; right thalamus; left lobule VIII of cerebellar hemisphere; and lobules IV, V, and VI of the vermis. The sensorimotor network was the densest part of the brain at rest, with the most hubs (mostly provincial) compared with the other networks. In contrast, the hubs in default mode, limbic, and cerebellar networks were mainly connectors.

### 4.4. Correlation Analysis

Correlation analysis indicates that more evening-oriented individuals (or late chronotypes) show lower degree centrality and nodal efficiency in the right hippocampus, lower nodal clustering coefficient and local efficiency in the right parahippocampal area, and higher degree centrality and nodal efficiency in right pallidum if examined in the morning, and shorter nodal paths in the left parahippocampal area, higher degree centrality in the right pallidum, and higher global assortativity in the evening. This may be interpreted, in a simplified way, so that later chronotype is associated with lower effectiveness of information transmission in the hippocampus and parahippocampal area during morning hours, while the transmission in the pallidum seems to be enhanced—both in the morning and evening. The morningness–eveningness dimension of chronotype refers to diurnal preferences and awareness of own performance level. In this context, lowered information flow in some structures in morning hours may be seen as a key indicator of eveningness. The pallidum node is more challenging to interpret. However, if one pays attention to the hedonic aspects of pallidum functions, it may be interesting to consider it in the context of individual differences in reward system sensitivity. Some research, applying various methodologies, suggest that evening types may be better “equipped” for processing pleasure and reward, e.g., Hasler et al. (2017) found a greater ventral striatum response to winning in young male evening-oriented individuals [105], while results of the cortical thickness analysis of Rosenberg et al. (2018) revealed greater grey matter volumes for late chronotypes in the left anterior insula [106]. Additionally, Norbury (2020) indicated (in a group of older adults) that self-reported eveningness was associated with increased grey matter volume in brain regions implicated in risk and reward processing (bilateral nucleus accumbens, caudate, putamen, and thalamus) and orbitofrontal cortex [107]. Finally, higher global assortativity linked with eveningness and manifested in evening hours may be seen as indirect proof of the accuracy of the subjective ME scale.

More distinct or “stronger” chronotypes (described by higher scores in the AM scale) tend to show lower global clustering coefficient, network local efficiency, and average path length as well as higher degree centrality and nodal efficiency in left precentral and postcentral areas, shorter nodal paths in left precentral and postcentral regions, and lower degree centrality and betweenness centrality in the right cerebellum in the morning, but lower nodal clustering coefficient and local efficiency in the right hippocampus as well as lower nodal local efficiency in the left parahippocampal area in the evening. These results indicate that strong chronotype is associated with effective information transmission in general sensing areas and less effective information transmission in the cerebellum in morning hours and lowered ability of specialized processing in hippocampal and parahippocampal areas in the evening. Subjective circadian amplitude is a complex construct referring to the range of diurnal variations of arousal, reflected in the strength of morning–evening preferences, flexibility, and stability of the rhythm [40]. Without a doubt, diurnal arousal changes indicate emotional lability and may be associated with emotional responsiveness and general sensitivity. Thus, the enhanced information transmission/flow in general sensing areas (precentral and postcentral) seems to be a logical correlate of large diurnal amplitude.

## 5. Conclusions

In the present study, we employed chronotype-based paradigms and performed graph-theory based network analysis in resting-state functional MRI to explore the topological differences in whole-brain functional networks between morning and evening sessions. The study results revealed meaningful information about the topological alterations of the brain network during the day. The results showed the effect of time of day on the functional connectivity patterns, but with no significant difference in chronotype categories. The chronotype-based paradigm is considered a highly sensitive tool for controlling circadian and homeostatic parameters [5]. The lack of differences between the topological alterations of the brain network during the day in the group of morning and evening-types suggests a universal character of the described phenomenon.

## 6. Limitations and Future Work

Several limitations in the current study should be considered for future directions. The first limitation concerns the atlas used in this study. We applied the AAL atlas to define 116 (cortical and subcortical) graph nodes for brain network construction. Although there is no consensus on which atlas is optimal for brain parcellation [108], some neuroscientists believe that the AAL atlas leads to inefficient parcel homogeneity [109]. A recommended atlas for handling this issue in future work is the cortical Schaefer/Yeo atlas [110]. Among the many advantages of the Schaefer/Yeo atlas is that each node is preassigned to a system based on a cross-validated study. Yet another limitation concerns the thresholding of the functional matrices. In fact, while thresholding does control for differences in binary density across subjects, it does not mean that the thresholded networks are representative of a given subject. Finally, comparing to the *t* and *F* tests, nonparametric permutation tests provide a more flexible and intuitive method for analyzing the data from functional neuroimaging studies [111]. Applying permutation tests which allow inferences to be made without prior assumptions should be considered in future studies.

## Figures and Tables

**Figure 1 brainsci-11-00111-f001:**
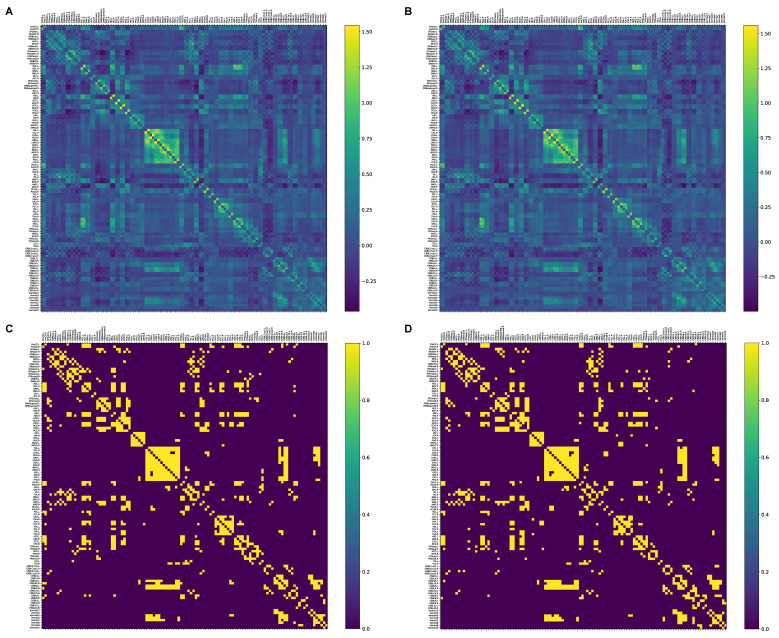
Correlation matrices (**A**,**B**) (transformed Fisher’s r-to-z) and 10% binarized matrices (**C**,**D**) for morning and evening sessions, respectively (averaged across all participants in each session). See Appendix A
Table A1 for the description of the areas.

**Figure 2 brainsci-11-00111-f002:**
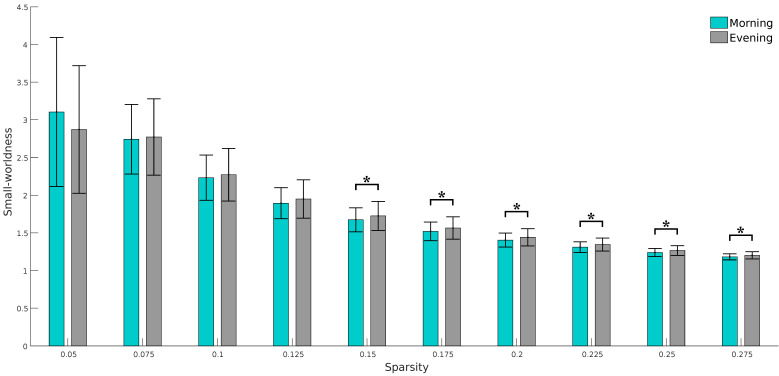
Results of paired *t*-test on the small-worldness at the threshold values of 0.05 to 0.275. Asterisks (*) in the figures show a significant difference in small-worldness between sessions (*p* < 0.01, Bonferroni corrected).

**Figure 3 brainsci-11-00111-f003:**
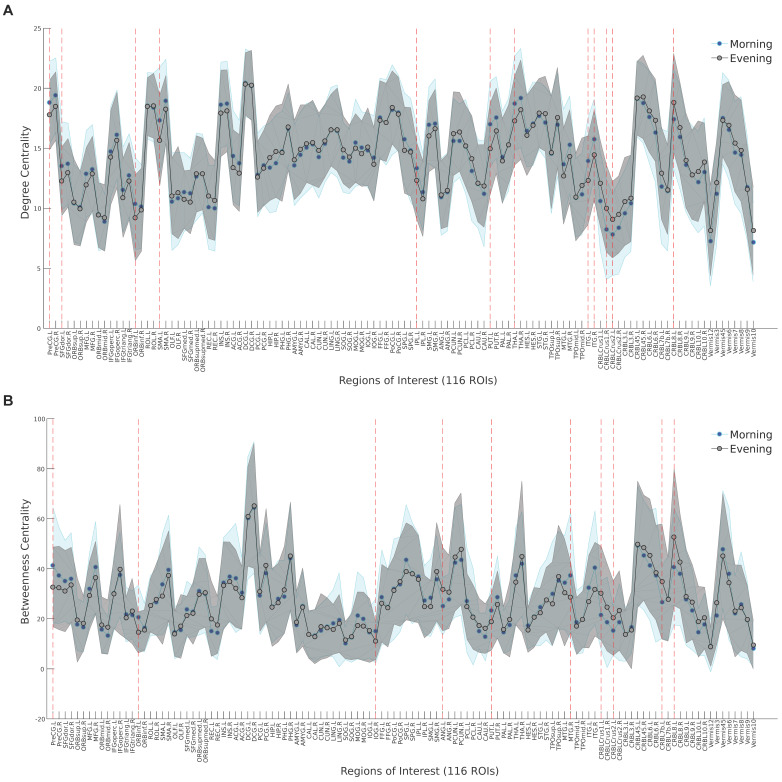
Area under the curve in the morning session (blue) and the evening session (gray) for degree centrality (**A**) and betweenness centrality (**B**) of all 116 brain regions. Significant diurnal fluctuations are represented by red lines. See Table A1 for the description of the areas.

**Table 1 brainsci-11-00111-t001:** Demographics, questionnaires, and actigraphy results.

Variables (Mean ± SD)	MT (*n* = 31)	ET (*n* = 31)	Significance
Sex (M/F) ^a^	11/20	12/19	Χ^2^(1) = 0.069; *p* = 0.793
Age (years) ^b^	24.45 ± 3.83	23.48 ± 2.55	U(62) = 446; *p* = 0.623
ME ^b^	15.71 ± 2.41	28.45 ± 2.39	U(62) < 0.001; *p* < 0.001
AM ^b^	21.47 ± 3.58	22.26 ± 3.51	U(62) = 426; *p* = 0.437
ESS ^b^	5.52 ± 2.48	5.87 ± 3.01	U(62) = 441; *p* = 0.576
EHI ^b^	86.83 ± 12.92	89.19 ± 13.93	U(62) = 414; *p* = 0.330
VNTR of PER3	5/5	4/4	-
Declared waketime (hh:mm) ^c^	07:07 ± 62 min	07:25 ± 48 min	t(60) = −1.90; *p* = 0.062
Declared bedtime (hh:mm) ^c^	23:24 ± 55 min	00:06 ± 49 min	t(60) = −3.50; *p* = 0.001
Declared length of perfect sleep (hh:mm) ^c^	08:50 ± 42 min	08:38 ± 54 min	t(60) = 1.54; *p* = 0.128
Actigraphy-derived waketime (hh:mm) ^c^	07:43 ± 70 min	08:16 ± 69 min	t(60) = −1.28; *p* = 0.168
Actigraphy-derived bedtime (hh:mm) ^c^	23:58 ± 58 min	00:48 ± 58 min	t(60) = −3.13; *p* = 0.002
Actigraphy-derived length of real sleep (hh:mm) ^c^	07:53 ± 51 min	07:36 ± 40 min	t(60) = −1.18; *p* = 0.266

MT—morning types, ET—evening types, ME—morningness/eveningness scale (Chronotype Questionnaire), AM—amplitude scale (Chronotype Questionnaire), ESS—Epworth Sleepiness Scale, EHI—Epworth Handedness Inventory, ^a^ chi-square test, ^b^ Mann–Whitney U test, ^c^ Student’s *t*-test.

**Table 2 brainsci-11-00111-t002:** List of brain regions of interest (ROIs) that changed throughout the day (significance level was set at *p* < 0.01 and *p*-values were adjusted for the Bonferroni correction).

ROI (Network)	MNI Coordinates			AAL Label	*p*-Value			
	x	y	z		DC	BC	CC	NE
1 (DMN)	−38.65	−5.68	50.94	Precentral_L	0.00043	0.00021		0.00024
3 (DMN)	−18.45	34.81	42.20	Frontal_Sup_L	0.00040			0.00045
15 (FPN)	−35.98	30.71	−12.11	Frontal_Inf_Orb_L	0.00043	0.00003		0.00025
19 (SMN)	−5.32	4.85	61.38	Supp_Motor_Area_L	0.00015			0.00010
54 (VN)	38.16	−81.99	−7.61	Occipital_Inf_R		0.00030		
61 (FPN)	−42.80	−45.82	46.74	Parietal_Inf_L	0.00042			0.00042
63 (SMN)	−55.79	−33.64	30.45	SupraMarginal_L	0.00065			0.00044
65 (FPN)	−44.14	−60.82	35.59	Angular_L		0.00004		
73 (LS)	−23.91	3.86	2.40	Putamen_L	0.00002	0.00043		0.00002
77 (LS)	−10.85	−17.56	7.98	Thalamus_L	0.00043			0.00046
78 (LS)	13.00	−17.55	8.09	Thalamus_R			0.00014	
83 (FPN)	−39.88	15.14	−20.18	Temporal_Pole_Sup_L			0.00033	
86 (DMN)	57.47	−37.23	−1.47	Temporal_Mid_R		0.00025		
89 (VN)	−49.77	−28.05	−23.17	Temporal_Inf_L	0.00015			0.00011
90 (VN)	53.69	−31.07	−22.32	Temporal_Inf_R	0.00028	0.00060		0.00029
91 (CRB)	−35.00	−67.00	−29.00	Cerebellum_Crus1_L	0.00066	0.00006		0.00069
92 (CRB)	38.00	−67.00	−30.00	Cerebellum_Crus1_R	0.00011	0.00081		0.00008
93 (CRB)	−28.00	−73.00	−38.00	Cerebellum_Crus2_L	0.00033	0.00037		0.00022
94 (CRB)	33.00	−69.00	−40.00	Cerebellum_Crus2_R	0.00096			0.00073
101 (CRB)	−31.00	−60.00	−45.00	Cerebellum_7b_L		0.00035		
103 (CRB)	−25.00	−55.00	−48.00	Cerebellum_8_L	0.00039	0.00006		0.00059
107 (CRB)	−22.00	−34.00	−42.00	Cerebellum_10_L		0.00052		

DC—Degree Centrality, BC—Betweenness Centrality, CC—Clustering Coefficient, NE—Nodal Efficiency.

**Table 3 brainsci-11-00111-t003:** Hub regions in different brain networks (at a sparsity of 0.1).

Network	Morning		Evening	
	L	R	L	R
Sensorimotor	Rolandic_Oper_L ^P^	Rolandic_Oper_R ^P^	Rolandic_Oper_L ^P^	Rolandic_Oper_R ^P^
	Insula_L ^P^	Insula_R ^P^	Insula_L ^P^	Insula_R ^P^
	Postcentral_L ^P^	Postcentral_R ^P^	Postcentral_L ^P^	Postcentral_R ^P^
	SupraMarginal_L ^P^	SupraMarginal_R^C^	-	SupraMarginal_R^C^
	Temporal_Sup_L ^P^	Temporal_Sup_R ^P^	Temporal_Sup_L ^P^	Temporal_Sup_R ^P^
Visual	Lingual_L ^P^	Lingual_R ^P^	Lingual_L ^P^	Lingual_R ^P^
	Fusiform_L ^P^	Fusiform_R ^P^	Fusiform_L ^P^	Fusiform_R ^P^
Frontoparietal	-	-	-	Temporal_Pole_Sup_R ^C^
Default Mode	Precentral_L ^C^	Precentral_R ^C^	Precentral_L ^C^	Precentral_R ^C^
Limbic	Cingulum_Mid_L ^C^	Cingulum_Mid_R ^C^	Cingulum_Mid_L ^C^	Cingulum_Mid_R ^C^
	-	Thalamus_R ^P^	-	-
Cerebellar	Cerebellum_4_5_L ^C^	Cerebellum_4_5_R ^C^	Cerebellum_4_5_L ^C^	Cerebellum_4_5_R ^P^
	Cerebellum_6_L ^C^	Cerebellum_6_R ^C^	Cerebellum_6_L ^C^	Cerebellum_6_R ^C^
	Vermis_4_5 ^P^	-	Cerebellum_8_L ^C^	-
			Vermis_6 ^P^	-

L/R—Left or Right Hemisphere, P—Provincial, C—Connector.

**Table 4 brainsci-11-00111-t004:** Partial correlations between nodal metrics with ME, AM, and ESS scores (*n* = 62; significance level was set at *p* < 0.01 and *p*-values were adjusted for the Bonferroni correction).

	ROI	Local Metrics	Partial Correlation (*p*-Value)		
			ME	AM	ESS
Morning	Hippocampus_R	Degree Centrality	−0.408 (0.0020)	-	-
		Nodal Efficiency	−0.361 (0.0080)	-	-
	ParaHippocampal_R	Nodal Clustering Coefficient	−0.367 (0.0068)	-	-
		Nodal Local Efficiency	−0.374 (0.0054)	-	-
	Pallidum_R	Degree Centrality	0.424 (0.0010)	-	-
		Nodal Efficiency	0.445 (0.0006)	-	-
	Precentral_L	Degree Centrality	-	0.361 (0.0080)	-
		Nodal Efficiency	-	0.402 (0.0024)	-
		Nodal Shortest Path	-	−0.465 (0.0002)	-
	Postcentral_L	Degree Centrality	-	0.395 (0.0030)	0.388 (0.0036)
		Nodal Efficiency	-	0.407 (0.0020)	0.358 (0.0086)
		Nodal Shortest Path	-	−0.410 (0.0018)	-
	Cerebellum_10_R	Degree Centrality	-	−0.378 (0.0048)	-
		Betweenness Centrality	-	−0.377 (0.0050)	-
Evening	Hippocampus_R	Nodal Clustering Coefficient	-	−0.440 (0.0006)	-
		Nodal Local Efficiency	-	−0.467 (0.0002)	-
	ParaHippocampal_L	Nodal Local Efficiency	-	−0.356 (0.0092)	-
		Nodal Shortest Path	0.382 (0.0044)	-	-
	Pallidum_R	Degree Centrality	0.353 (0.0098)	-	-

ME—Morningness/Eveningness Scale, AM—Amplitude Scale, ESS—Epworth Sleepiness Scale.

## Data Availability

The experimental fMRI data are available with the correspondence M.F., vonfrovitz@gmail.com; magda.fafrowicz@uj.edu.pl.

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
