# Peer review of "Identifying Diurnal Variability of Brain Connectivity Patterns Using Graph Theory"

_brainsci, 2021, doi:10.3390/brainsci11010111_

Round 1
Reviewer 1 Report
The manuscript, “Identifying Diurnal Variability of Brain Connectivity Patterns using Graph Theory”, discusses the research results concluding time-of-day effect connectivity profiles but not for chronotypes, i.e. individuals who exhibit their highest alertness and energy level either in the early morning or late evening. These effects were exhibited predominantly across the left hemisphere, in areas such as the precentral gyrus, putamen, inferior frontal gyrus (orbital part), inferior temporal gyrus, as well as the bilateral cerebellum.
The manuscript was very well written and the analysis and results leading up to the discussion was very well formulated. There are several areas that need some work:
Introduction:
Lines 48-50 state, “several studies have been conducted to identify topological changes in the brain networks that help us better understand the mechanisms underlying human cognition and neurological disorders [26–35]”. While the authors cite the instances, it would strengthen the authors case to elaborate 2-3 of the studies.
Results:
Figure 2 – The authors mention several tools for their analysis, i.e. Global properties such as global efficiency, modularity, average shortest path, small-worldness, and assortativity can be used to provide global information flow and functional specialization.
How was small-worldness used to develop Figure 2 – please elaborate. Why were these thresholds selected?
Discussion:
Lines 420-422 state, However, if one puts attention on hedonic aspects of pallidum functions, it may be interesting to consider it in the context of individual differences in reward system sensitivity. Some research suggest that evening types may be better ‘equipped’ for processing of hedonic experiences – Norbury (2020).
It is a stretch to interpret that Norbury considered hedonic aspects of pallidum functions in the elderly – consider other references or re-interpret the statement.
Author Response
Response to Reviewer 1 Comments
Introduction:
- Lines 48-50 state, “several studies have been conducted to identify topological changes in the brain networks that help us better understand the mechanisms underlying human cognition and neurological disorders [26–35]”. While the authors cite the instances, it would strengthen the authors case to elaborate 2-3 of the studies.
Page 2, lines 60-67: We elaborated several studies in more details of their findings as: “For example, Lunsford-Avery et al. (2020) studied the relation between the regularity of sleep/wake patterns and brain connectome among adolescents and young adults to measure how these naturalistic patterns contribute to default mode network (DMN) topology [31]. In another study, Farahani et al. (2019) examined the effects of sleep restriction on the brain functional network and found significant topological alterations mostly across the limbic system, DMN, and visual network [33]. Disrupted brain network topology was examined in studies on patients with Parkinson's disease [32], chronic insomnia [36], major depressive disorder [39], as well as preterm infants [35].”
Results:
- Figure 2 – The authors mention several tools for their analysis, i.e. Global properties such as global efficiency, modularity, average shortest path, small-worldness, and assortativity can be used to provide global information flow and functional specialization.
How was small-worldness used to develop Figure 2? Please elaborate. Why were these thresholds selected?
Page 4, lines 159-162: We explained is as: “All measures were extracted from the thresholded and binarized networks with the sparsity between 0.05 and 0.275 (step of 0.025). The reason for choosing this interval was to reduce computational complexity, while preventing the creation of disconnected graphs.” Also, small-worldness is computed as the ratio of normalized clustering coefficient to normalized path length.
Discussion:
- Lines 420-422 state, however, if one puts attention on hedonic aspects of pallidum functions, it may be interesting to consider it in the context of individual differences in reward system sensitivity. Some research suggest that evening types may be better ‘equipped’ for processing of hedonic experiences – Norbury (2020).
It is a stretch to interpret that Norbury considered hedonic aspects of pallidum functions in the elderly – consider other references or re-interpret the statement.
Page 13, lines 465-475: We explained and interpreted the comment as: “Some research, applying various methodologies, suggest that evening types may be better ‘equipped’ for processing pleasure and reward, e.g., Hasler et al. (2017) found a greater ventral striatum response to win in young male evening-oriented individuals [105], while Rosenberg’s et al. (2018) results of cortical thickness analysis revealed greater grey matter volumes for late chronotypes in the left anterior insula [106]. Also, Norbury (2020) indi-cated (in the group of older adults) that self-reported eveningness was associated with in-creased grey matter volume in brain regions implicated in risk and reward processing (bilateral nucleus accumbens, caudate, putamen and thalamus) and orbitofrontal cortex [107]. Finally, higher global assortativity linked with eveningness and manifested in evening hours may be seen as an indirect proof of the accuracy of subjective ME scale.”
Reviewer 2 Report
Title: Identifying Diurnal Variability of Brain Connectivity Patterns using Graph Theory
Journal: Brain Sciences
Abstract
- Authors should add the number of males and females as well as the mean and standard deviation of age of the overall sample.
- Authors should explain the functional meaning of “higher small-worldness in the evening session”.
Introduction
- Page 1, line 39. It seems to me that authors missed to describe the intermediate or neither types.
- Page 1, line 39. Authors should quote the last comprehensive review published on circadian typology by Adan and colleagues: https://pubmed.ncbi.nlm.nih.gov/23004349/
- Page 1, line 40. Authors should replace “topology” with “typology”.
- Page 1, lines 43-45. Authors wrote “Although effects of circadian rhythms and chronotypes on whole-brain connectivity have been examined, the results are often contradictory and inconsistent; in many cases, only one of the two factors has been considered [11–14]”. Authors should summarize the findings of previous-similar studies.
- Page 2, line 58. Authors should add the expected results.
Materials and Methods
- Page 2, line 64. Authors should replace “subjects” with “participants”.
- Page 2, lines 77-78. Authors should add the ethical committee report number.
- Table 1. Authors should add the complete statistics referred to each group comparison.
Results
- Table 2. Since p-values are adjusted for the Bonferroni correction, I expected that the significance level was set to a value lower than p<.05.
- Table 4. Please see my previous comment.
Discussion
- Authors should provide a detailed explanation of the reason why they did not observe any synchrony effects.
- Page 9, lines 207-217. Which is the functional meaning of these findings?
- Page 9, line 230. Authors should quote the following review on sleep inertia: https://pubmed.ncbi.nlm.nih.gov/31692489/
Author Response
Response to Reviewer 2 Comments
Abstract
- Authors should add the number of males and females as well as the mean and standard deviation of age of the overall sample.
Page 1, line 20: We completed the related sentence in the abstract as: “In this regard, 62 participants (39 women; mean age: 23.97 ± 3.26 years; half morning- versus half evening-type) were scanned about 1 hour and 10 hours after wake-up time for morning and evening sessions, respectively.”
- Authors should explain the functional meaning of “higher small-worldness in the evening session”.
Page 1, lines 23-26: We explained the meaning of small-worldness as: “Compared with the morning session, we found relatively higher small-worldness (an index that represents more efficient network organization) in the evening session, which suggests the dominance of sleep inertia over the circadian and homeostatic processes in the first hours after waking.”
Introduction
- Page 1, line 39. It seems to me that authors missed to describe the intermediate or neither types.
We cleared the sentence as: “Accordingly, people can be divided into morning-type (or early larks), evening-type (or night owls), and intermediate-type (or neither-type) [12].”
- Page 1, line 39. Authors should quote the last comprehensive review published on circadian typology by Adan and colleagues: https://pubmed.ncbi.nlm.nih.gov/23004349/
Great! We cited the recommended paper in the sentence of: “Chronotype differences have been reported to be highly influential on people’s cognition, behavior, and daily neural activity [1,5,12,14–16].”
- Page 1, line 40. Authors should replace “topology” with “typology”.
We replaced “topology” with “typology”.
- Page 1, lines 43-45. Authors wrote “Although effects of circadian rhythms and chronotypes on whole-brain connectivity have been examined, the results are often contradictory and inconsistent; in many cases, only one of the two factors has been considered [11–14]”. Authors should summarize the findings of previous-similar studies.
Page 2, lines 46-54: We summarize the results of the previous studies in that paragraph as follows: “Although effects of circadian rhythms and chronotypes on whole-brain connectivity have been examined (some cases have only considered time of day [17]), the results are of-ten contradictory and inconsistent. For example, a group of researchers believe that rest-ing-state brain networks maintain a constant topological organization throughout the day [18,19], while others believe that brain networks, especially default mode, sensorimotor, and visual networks, show significant changes as the day progressed when we are at rest [20–22]. Also, Orban et al. [23], contrary to common belief that global brain signal is low in the morning and then increases in the midafternoon, and drops in the early evening, showed that the global signal fluctuation is continuously decreasing during the day.”
- Page 2, line 58. Authors should add the expected results.
Page 2, lines 70-72: We added a sentence explaining the expected results as: “Based on our previous findings, we hypothesized that topological changes were mostly because of time of day rather than chronotype, and areas such as the default mode and sensorimotor networks underwent the most changes.”
Materials and Methods
- Page 2, line 64. Authors should replace “subjects” with “participants”.
We replaced “subjects” with “participants”.
- Page 2, lines 77-78. Authors should add the ethical committee report number.
We added the details of study approval by a properly constituted research ethics committees at the end of the Participants section.
- Table 1. Authors should add the complete statistics referred to each group comparison.
We added the statistics regarding the group comparison in Table 1.
Results
- Table 2. Since p-values are adjusted for the Bonferroni correction, I expected that the significance level was set to a value lower than p<.05.
- Table 4. Please see my previous comment.
For both Tables 2 and 4, we converted all the p-values and significant levels to the original Bonferroni correction format. The significance levels for global analysis and correlation analysis were set to 0.01 and the significance level for local analysis was set to 0.001 (the previous significance level was considered only to normalize all the p-values for all analysis (global, local, and correlation) in the same scale by a correspondent multiplication, however, the values were controlled for FDR).
Discussion
- Authors should provide a detailed explanation of the reason why they did not observe any synchrony effects.
Page 8, lines 242-248: We explained this in the discussion (first paragraph) as: “The effect of chronotype and interaction between time-of-day and chronotype (so-called synchrony effect) were not observed in global and local analyses, which is in line with our previous study [48]. The synchrony effect was confirmed in a variety of cognitive domains [5,49] and in task fMRI characterized by high complexity [15]. In relation to the resting-state data, some recent reports revealed the influence of chronotype on resting-state functional connectivity (with contradicting results) [50,51], however, they did not confirm the synchrony effect.”
- Page 9, lines 207-217. Which is the functional meaning of these findings?
Page 8, lines 248-251: We added a sentences in the discussion as: “Our findings regarding global and local connectivity profiles indicate the variability of brain’s functional organization between morning and evening resting-state sessions as a universal phenomenon, independent of circadian typology.”
- Page 9, line 230. Authors should quote the following review on sleep inertia: https://pubmed.ncbi.nlm.nih.gov/31692489/
Page 9, lines 274-275: We cited the recommended paper as: “However, the sleep inertia’s exact function as well as its neurophysiological basis are still not fully known [for a review see: 56].”